# Role of the Anaphase-Promoting Complex Activator Cdh1 in the Virulence of *Cryptococcus neoformans*

**DOI:** 10.3390/jof10120891

**Published:** 2024-12-23

**Authors:** Qiu-Hong Liao, Lian-Tao Han, Meng-Ru Guo, Cheng-Li Fan, Tong-Bao Liu

**Affiliations:** 1State Key Laboratory of Resource Insects, Southwest University, Chongqing 400715, China; lqh1994@email.swu.edu.cn (Q.-H.L.); guomr1997@email.swu.edu.cn (M.-R.G.); 2Medical Research Institute, Southwest University, Chongqing 400715, China; hlt892996713@email.swu.edu.cn; 3College of Animal Science and Technology, Southwest University, Chongqing 400715, China; 4Jinfeng Laboratory, Chongqing 401329, China; 5Engineering Research Center for Cancer Biomedical and Translational Medicine, Southwest University, Chongqing 400715, China

**Keywords:** *Cryptococcus neoformans*, APC activator Cdh1, capsule, virulence, chemokines

## Abstract

*Cryptococcus neoformans* is a globally distributed human fungal pathogen that can cause cryptococcal meningitis with high morbidity and mortality. In this study, we identified an anaphase-promoting complex (APC) activator, Cdh1, and examined its impact on the virulence of *C. neoformans*. Our subcellular localization analysis revealed that Cdh1 is situated in the nucleus of *C. neoformans*. Disrupting or overexpressing the *CDH1* gene caused abnormal capsule formation in *C. neoformans*. The *cdh1*Δ mutant displayed slight sensitivity when grown at 37 °C, indicating that Cdh1 plays a role in maintaining the growth of *C. neoformans* at 37 °C. A fungal virulence assay showed that Cdh1 is closely associated with the virulence of *C. neoformans*, and both the *cdh1*Δ mutant and *CDH1*^OE^ overexpression strains significantly diminished the virulence of *C. neoformans*. The *Cryptococcus*–macrophage interaction assay revealed that both the *cdh1*∆ mutant and the *CDH1*^OE^ strains had significantly lower proliferation ability inside macrophages. Furthermore, the infection of the *cdh1*Δ mutant significantly activated neutrophil recruitment, as well as Th2 and Th17 immune responses, in lung tissue. In summary, our findings indicate that Cdh1 is crucial for producing virulence factors and fungal virulence in *C. neoformans*. The findings of this study can offer valuable insights and form the basis for further study of the regulatory mechanisms governing the pathogenicity of *C. neoformans*, potentially leading to the development of novel therapeutic strategies.

## 1. Introduction

*Cryptococcus neoformans* is an opportunistic fungal pathogen found widely in the natural environment. The desiccated spores or yeast cells from the environment can invade the human body through the respiratory tract, causing cryptococcal pneumonia. This initial infection can further disseminate via the bloodstream, resulting in systemic infections involving various organs such as skin, eyes, and bones [1,2,3,4]. Moreover, it can infiltrate the central nervous system, causing a fatal infection, cryptococcal meningitis [5]. It is estimated that *C. neoformans* causes about 194,000 infections and 147,000 deaths each year [6]. In 2022, the World Health Organization identified *C. neoformans* as a critical-priority fungal pathogen, posing a significant threat to public health worldwide [7].

As a pathogenic fungus, *C. neoformans* possesses a variety of classical virulence factors, such as the ability to survive at 37 °C, the production of polysaccharide capsules, and the formation of melanin. The primary prerequisite for cryptococcal infection of mammalian hosts is the ability to grow at 37 °C. The presence of polysaccharide capsules can inhibit the phagocytosis of *C. neoformans* by host immune cells, and the phagocytosis efficiency of immune cells decreases with an increase in capsule thickness [8,9]. Conversely, the mutant strains with an impaired capsule structure are often more susceptible to phagocytosis by macrophages [10,11,12]. The shed capsule antigen can also cause damage to the host immune system through various mechanisms [13]. Melanin can protect pathogenic fungi from phagocytosis by pulmonary macrophages by eliminating antimicrobial oxides produced in phagocytic cells, reducing host-killing and immune response ability [14]. Melanin can enhance the nonlytic exocytosis of *C. neoformans*, allowing it to evade phagocytic cells and decreasing the effective concentration of drugs by binding them in the cell wall, ultimately facilitating the dissemination of *C. neoformans* [15,16]. In addition, *C. neoformans* can also produce virulence factors such as urease [17], phospholipase [18,19,20], and protease [21,22]. The production of virulence factors helps *C. neoformans* adapt to the host environment and resist host defense responses against pathogens.

Cyclins are a class of proteins that play a crucial role in regulating the cell cycle, a process essential for maintaining genomic integrity in organisms. The dysregulation of cyclin activity has been implicated in the development of various types of cancer [23], such as breast cancer [24], myeloma [25], colon cancer [26], ovarian cancer [27], and oral cancer [28]. Additionally, cell cycle regulatory proteins have been found to contribute to the pathogenicity of human fungal pathogens [29]. For example, in *C. neoformans*, the G1 cyclin gene *CLN1* has been identified, and its mRNA has been shown to be periodically expressed during the cell cycle [30]. Mutant strains lacking the G1/S cyclin *CLN1* exhibit enlarged capsules, growth defects at higher temperatures, and an inability to synthesize melanin and capsules, ultimately resulting in a loss of pathogenicity [31,32]. Similarly, the deletion of the cyclin Cbc1 in *C. neoformans* significantly reduces the strain’s ability to produce virulence factors, leading to a complete loss of pathogenicity [33]. These findings underscore the critical role of cell cycle-related proteins in the evolution of virulence in fungal pathogens.

*C. neoformans* is an important fungal pathogen that poses a substantial threat to human and public health. The complexity inherent in the structure of fungal cell walls presents significant challenges in the treatment of fungal infections, rendering them more difficult to manage than bacterial and viral infections. Moreover, the available antifungal drugs are limited and expensive and face limitations in efficacy due to drug resistance, toxicity, and high treatment costs. This results in significant challenges in managing fungal infections [34,35], leading to high morbidity and mortality rates associated with conditions such as cryptococcosis. Understanding the pathogenic mechanism of *C. neoformans* has significant implications for developing new drugs and effectively treating cryptococcosis. In this study, we identified an anaphase-promoting complex activator, Cdh1, and investigated its role in the virulence of *C. neoformans*. Our findings indicate that Cdh1 significantly influences the formation of virulence factors and plays a crucial role in the virulence of *C. neoformans*.

## 2. Materials and Methods

### 2.1. Strains and Media

The strains used in this study are detailed in Appendix A [36,37]. *E. coli* DH5α was grown in a lysogeny broth (LB, 0.5% yeast extract, 1% tryptone, and 1% NaCl). *C. neoformans* strains were cultured on a YPD medium (1% yeast extract, 2% peptone, and 2% dextrose). Melanin formation was induced using a Niger seed medium, and capsule formation in *C. neoformans* was induced by culturing in a Sabouraud medium diluted with 50 mM MOPS at pH 7.3 [38]. All other media used in this study were prepared as previously described [39]. The strains were stored as 20% glycerol stocks at −80 °C.

### 2.2. Construction of Recombinant Strains

The plasmids and primers used in this study are shown in Appendix A [40,41,42,43] and Appendix A. Primer synthesis and sequence determination were carried out by Sangon Biotech (Sangon Biotech, Shanghai, China). Plasmid construction was performed by homologous recombination according to the instructions of the Hieff Clone^®^ Plus One Step Cloning Kit (Yeasen, Shanghai, China). All the restriction enzymes used in this study were obtained from New England Biolabs (NEB, Beijing, China). PCR amplification was performed using the Hieff Canace^®^ Gold High Fidelity DNA Polymerase (Yeasen, Shanghai, China). *C. neoformans CDH1*-related strains were obtained through biolistic transformation, as described previously [44].

The construction of the *cdh1*Δ mutant strain involved the targeted replacement of the *CDH1* gene with the *NEO* resistance gene through homologous recombination. Initially, the upstream (841 bp) and downstream (820 bp) flanking fragments of *CDH1* were amplified from the wild-type strain H99 genome using the TL1828/TL1829 and TL1830/TL1831 primers. Simultaneously, the *NEO* marker gene (2098 bp) was amplified using TL17/TL18 as the primers and pJAF1 as the template. Subsequently, overlap PCR was conducted to fuse the *CDH1* upstream with *NEO* (2194 bp) using the primer TL1828/TL20. The downstream fragment of *CDH1* was fused to *NEO* (1823 bp) using the primer TL19/TL1831. The resulting fusion fragments were purified and concentrated to 2 μg/μL, followed by transformation into α or **a** mating type of wild-type *C. neoformans* cells using biolistic transformation. The *cdh1*Δ mutants were selected using 200 mg/L G418 (Sangon Biotech, Shanghai, China) and further confirmed via Southern blot analysis using the DIG-High Prime DNA Labeling and Detection Starter Kit II (Roche, Mannheim, Germany). The successfully constructed α and **a** mating type *cdh1*Δ mutant strains were respectively designated as TBL508 and TBL509.

To obtain a complemented strain of the *cdh1*Δ mutant, a 4.1 kb fragment of the wild-type strain H99 genome was amplified with primers TL1947/TL1948 and was subsequently cloned into the *Bam*HI site of pTBL1 to generate the recombinant plasmid pTBL404. This fragment contained a segment of approximately 0.8 kb upstream and a terminator segment of approximately 0.9 kb downstream of the coding region of the *CDH1* gene. The linearized pTBL404, following digestion with *Xho*I, was purified and concentrated to 2 μg/μL. The linearization plasmid pTBL404 was transformed into α or **a** mating type of *cdh1*Δ mutants by biolistic transformation. After screening for 100 mg/L NAT (Qingpu, Shanghai, China) resistance and confirmation via PCR using the primers TL1945/TL1946, the obtained *cdh1*::*CDH1* complemented strains of α and **a** mating types were designated as TBL512 and TBL513, respectively.

To generate a *CDH1* overexpression strain (hereafter called *CDH1*^OE^), the *CDH1* gene (2370 bp) was amplified from the wild-type H99 genome using the primers TL1945/TL2192 and subsequently cloned into the *Bam*HI site of pTBL405, yielding the *CDH1*: *MYC* fusion plasmid pTBL443 in which the expression of the *CDH1* gene was under the control of an action promoter. Following linearization with *Sca*I, the purified pTBL443 was concentrated to 2 μg/μL and introduced into *cdh1*Δ mutant strains via biolistic transformation. Transformants were screened for resistance to 150 mg/L hygromycin (Sangon Biotech, Shanghai, China), followed by total protein extraction and immunoblotting using a Myc-tag Polyclonal antibody (Proteintech Group, Wuhan, China) to detect the expression of Cdh1:Myc.

To evaluate the expression level of *Cryptococcus* genes, we conducted qRT-PCR to verify the transcription levels of the mRNA associated with *C. neoformans*. Overnight cultures of each *C. neoformans* strain were collected and washed twice with sterile ddH_2_O to obliterate the medium. Subsequently, the yeast cells were freeze-dried, and total RNA was extracted using the Omega Bio-Tek Total RNA Kit I (Norcross, GA, USA) following the manufacturer’s instructions. RNA quantification was performed using a MicroSpectrophotometer (K5600C; KAIAO, Shanghai, China), and cDNA was synthesized using the Hifair^®^ III 1st Strand cDNA Synthesis SuperMix for qPCR (Yeasen, Shanghai, China) with the primers TL2135/TL2136 (*CDH1*), TL2466/TL2467 (*APP1*), and TL217/TL218 (*GAPDH*) on the Light Cycler 96 QPCR system (Roche, Mannheim, Germany). Gene expression levels were normalized using the endogenous control gene *GAPDH*, and relative expression was determined using the 2^−ΔΔCT^ method [45]. Statistically significant differences were evaluated using Dunnett’s multiple comparison test (GraphPad Software Inc., San Diego, CA, USA). The *CDH1*^OE^ overexpression strains of the α and **a** mating type were named as TBL580 and TBL575, respectively.

To determine the subcellular localization of Cdh1 protein in *C. neoformans*, the coding region of the *CDH1* gene (2373 bp) was amplified by TL2119/TL2120 primers from the wild-type strain H99 genome. The amplified gene was cloned into pCN19 to construct the GFP-Cdh1 fusion protein expression vector, designated as pTBL435. Subsequently, the constructed plasmid was linearized with *Sac*I, concentrated to 2 µg/µL, and transformed into the *cdh1*Δ mutant strain via biolistic transformation. Transformants were initially screened on a YPD medium supplemented with nourseothricin sulfate (100 mg/L), and the fluorescence of the strain was further confirmed using a fluorescence microscope using a 100× objective lens (Axo observer 3; Zeiss, Oberkochen, Germany). The resulting Cdh1 green fluorescent strain was named TBL630.

### 2.3. Subcellular Localization Analysis of Cdh1 (DAPI Staining)

To determine the localization of Cdh1 in *C. neoformans*, we performed DAPI staining. Briefly, 500 μL of overnight cultures were collected and washed twice with 1 mL sterile 1 × PBS buffer (pH 7.4). Subsequently, 200 μL of 9.3% formaldehyde was added to fix the cells at room temperature for 10 min. Following fixation, the cells were washed twice with 1 mL 1 × PBS buffer. Then, the cells were resuspended in 300 μL 1 × PBS buffer, an equal volume of 1 × PBS buffer containing 1% Triton X-100 was added, and the permeabilization was allowed to proceed for 10 min at room temperature. Afterward, the reaction solution was washed twice with 1 mL of 1 × PBS buffer. Next, the cells were resuspended in 300 μL of PBS, after which we added an equal volume of 10 μg/mL DAPI (Sigma, Saint Louis, MO, USA) staining solution. After thorough mixing, the cells were incubated at 60 rpm for 15 min at room temperature. Finally, the cells were washed twice with 1 mL of PBST buffer, resuspended, and observed with a light microscope using a 100× objective lens (Axo observer 3; Zeiss, Oberkochen, Germany). To investigate the localization of the Cdh1 protein under different stress conditions, we inoculated the GFP-Cdh1 fluorescent strain into a YPD medium containing various stresses such as cell integrity stress (0.025% SDS and 0.5% Congo red), osmotic stress (1.5 M NaCl and 1.5 M sorbitol), and oxidative stress (2.5 mM H_2_O_2_). Subsequently, the subcellular localization was observed with fluorescence microscopy (Axo observer 3; Zeiss, Oberkochen, Germany) after 24 h.

### 2.4. Assays of Melanin and Capsule Production and Growth Under Stress Conditions

To assess melanin production in *C. neoformans* strains, 100 μL of each overnight culture was inoculated onto a Niger seed agar medium. The agar plates were then incubated at either 30 °C or 37 °C for 2 days, after which the pigmentation of the fungal colonies was evaluated and photographed for recording. For capsule production analysis, overnight cultures were washed three times with phosphate-buffered saline (PBS) and resuspended at a concentration of 1 × 10^6^ cells/mL in a DME medium. The cells were incubated in a 10% CO_2_ atmosphere for 24 h. After the capsule induction, the cells were mixed with India ink, observed by a light microscope using a 100× objective lens (Axo observer 3; Zeiss, Oberkochen, Germany), and photographed for recording. The capsule length of *C. neoformans* ((extracellular diameter-intracellular diameter)/2) was measured using Image J, and at least 100 cells were measured for each strain.

To investigate the role of Cdh1 in the stress resistance of *C. neoformans*, we examined the growth of each cryptococcal strain on YPD plates containing different stressors. Overnight cultures of each *C. neoformans* strain were collected, washed three times in sterile PBS, and diluted to an OD_600_ of 2.0. The yeast cells were serially diluted ten-fold in ddH_2_O, and 5 μL aliquots of each dilution were plated. The plates were incubated at 30 °C for 2–3 days, and the growth status of the strains was recorded by photographing.

To evaluate the role of Cdh1 in the growth of *Cryptococcus* under different temperatures, we adjusted the initial concentration of the inoculum to 1 × 10^7^ cells/mL and incubated 100 μL of the inoculum at 30 °C and 37 °C for 36 h, with measurements taken every 2 h. The optical density (OD_600_) was determined using the BMG LABTECH multifunctional microplate reader (FLUOstar Omega, Ortenberg, Germany).

### 2.5. Virulence Studies

In this study, overnight cultures of each cryptococcal strain were washed three times with 1 × PBS buffer and then resuspended at 2 × 10^6^ cells/mL in 1 × PBS buffer. Subsequently, ten female eight-week-old C57 BL/6 mice (Hunan SJA Laboratory Animal, Changsha, China) were intranasally inoculated with 50 μL of cell suspension of each strain, resulting in an inoculum of 1 × 10^5^ cells per mouse. The infected mice were regularly monitored after inoculation, and their survival times were recorded. The survival curve of the mice was then plotted using GraphPad Prism 8.0.

To further investigate the role of Cdh1 in the pathogenicity of *C. neoformans*, we conducted an assessment of the virulence associated with Cdh1-related strains using the Wax moth larva (*Galleria mellonella*) model, as outlined in previously published protocols [46]. Overnight cultures of each strain were harvested, subjected to two washes with sterile 1 × PBS buffer, and then adjusted to a concentration of 1 × 10^8^ cells/mL in 1 × PBS buffer. Groups of 10 larvae, with an average body weight of 300 mg, were inoculated with 10 μL of the inoculums with a microliter syringe, targeting the area of the last pro-leg. Prior to injection, the pro-leg area was disinfected with 70% ethanol using a swab. Following inoculation, the larvae were placed in 90 mm plastic Petri dishes and incubated at 30 °C and 37 °C in the dark for 16 days. During the incubation period, the larvae were monitored daily by observing spontaneous or provoked movements, facilitated by the use of a previously sterilized clip. The larvae were deemed deceased when they exhibited a lack of response to gentle prodding with forceps. Survival curves were generated, and the statistical analysis of survival differences was conducted employing the Kaplan–Meier method, utilizing GraphPad Prism 8.0.

All animal studies conducted at Southwest University were according to the “Guidelines on Ethical Treatment of Experimental Animals (2006, No. 398)” and the “Regulations for the Management of Laboratory Animals” (2006, No. 195) issued by the Ministry of Science and Technology of China. Furthermore, the Animal Ethics Committee at Southwest University approved all of the vertebrate studies under Protocol No. IACUC-20221022-14).

### 2.6. Histopathology and Fungal Burdens of Infected Tissues

The infected lungs, brains, and spleens of the mice were dissected, and half of the organs were fixed with 10% formalin for histopathological section preparation by Servicebio (Servicebio, Wuhan, China). The other half of the organs were homogenized in 3 mL of sterile 1 × PBS buffer, and 100 μL of the serial dilutions of the homogenates were plated on YPD agar plates containing 50 mg/L chloramphenicol and 50 mg/L kanamycin, followed by an incubation period of 2–3 days at 30 °C. The resulting colonies were counted, and the tissue fungal burdens were calculated as CFU/g tissue weight, represented as log10. Additionally, the histopathological section of infected tissues was stained with hematoxylin and eosin (H&E) and recorded using a light microscope with a 40× objective lens (Axo observer 3; Zeiss, Oberkochen, Germany).

### 2.7. Cryptococcus–Macrophage Interaction Assay

Macrophage-like J774 cells were cultured in Dulbecco’s Modified Eagle Medium (DMEM) supplemented with 10% heat-inactivated fetal bovine serum (FBS) at 37 °C under a 5% CO_2_ atmosphere. A total of 5 × 10^4^ J774 cells in 0.5 mL of fresh DMEM were seeded into each well of a 48-well culture plate and incubated overnight at 37 °C in 5% CO_2_. To facilitate the activation of macrophage cells, 50 units/mL of gamma interferon (IFN-γ; Invitrogen) and 1 µg/mL of lipopolysaccharide (LPS; Sigma) were administered to each well. *C. neoformans* overnight cultures were subsequently washed twice with phosphate-buffered saline (PBS) and opsonized using 10 µg/mL 18B7 mAb (Meridian Bioscience, Beijing, China). A total of 2 × 10^5^
*C. neoformans* cells were introduced to each well, achieving a yeast-to-J774 cell ratio of 4:1. To evaluate the intracellular proliferation of *C. neoformans*, non-adherent extracellular yeast cells were removed by washing with fresh DMEM following 2 h of co-incubation. The cultures were further incubated for additional time points of 0, 2, or 22 h. At these specified time intervals, the medium in each well was replaced with distilled water (dH_2_O) to lyse the macrophage cells for 30 min at room temperature. The resulting lysate was then spread onto YPD plates containing 50 mg/L chloramphenicol and 50 mg/L kanamycin and incubated at 30 °C for 2 days, after which colony-forming units (CFU) were counted to assess the intracellular proliferation of the organism.

Serum treatment and the *C. neoformans* cell viability assay were conducted as previously described [47]. At the specified time points, aliquots were removed and plated onto a YPD medium after serial dilution to assess cell viability.

### 2.8. Statistical Analysis

Statistical analysis and plotting of the data in this study were conducted using Prism 8.0. Significance was determined using a one-way ANOVA with Tukey’s multiple comparisons in GraphPad Prism. A log-rank test was utilized to analyze the differences in pathogenicity between mice or between the Wax moth larvae infected with different strains. All data are presented as mean ± standard deviation, and a *p*-value < 0.05 was considered statistically significant.

## 3. Results

### 3.1. Identification of the Anaphase-Promoting Complex Activator Cdh1 in C. neoformans

In our previous study, we identified the potential RNA targets associated with the RNA-binding protein Grp1. Our findings suggest that CNAG_03191 is a promising candidate gene associated with Grp1(data not published). We downloaded the sequence information of CNAG_03191 from the Fungi DB website (https://fungidb.org/fungidb/app/record/gene/CNAG_03191, accessed on 30 June 2022) and conducted a thorough analysis. The sequence analysis revealed that the full length of CNAG_03191 was 2726 bp, containing four exons and three introns (Figure 1A). Furthermore, domain analysis showed that CNAG_03191 contained six WD40 repeats (Figure 1B). The WD40 repeat is a short ~40 amino acid motif, usually terminated by a Trp-Asp (W-D) dipeptide, which is widespread and highly conserved in eukaryotes and usually folds into a β-propeller structure [48]. We searched for sequences of *S. cerevisiae* with high similarity to *C. neoformans* CNAG_03191 on NCBI and then compared and analyzed these sequences using the clustal W (https://www.genome.jp/tools-bin/clustalw/, accessed on 30 June 2022) and ESPript 3.0 (https://espript.ibcp.fr/ESPript/ESPript/, accessed on 30 June 2022) tools. Our results demonstrated that the amino acid sequence of *C. neoformans* CNAG_03191 shares 73.38% similarity (Figure 1C) with the *S. cerevisiae* anaphase-promoting complex activator Cdh1; so, we designated CNAG_03191 as Cdh1 based on this similarity.

### 3.2. Cdh1 Is Localized in the Nucleus of C. neoformans

To determine the subcellular localization of Cdh1 protein in *C. neoformans*, a *GFP*-*CDH1* fusion expression vector, pTBL435, was constructed and transformed into the *cdh1*Δ mutants using biolistic transformation. Following resistance screening and fluorescence microscopy observation, the GFP-Cdh1 green fluorescent strain was obtained. The fluorescence observation revealed that the GFP-Cdh1 fusion protein exhibited a punctate green fluorescence signal and was situated at the center of *C. neoformans* cells. Further confirmation of the localization of Cdh1 was achieved by staining cells with DAPI, which demonstrated the co-localization of Cdh1 with DAPI, indicating its presence in the nucleus (Figure 2A). Moreover, the expression of the GFP-Cdh1 fusion protein remained stable in the nucleus of *C. neoformans* under various stress conditions (Figure 2B).

### 3.3. Construction of Cdh1-Related Strains

To investigate the functional role of Cdh1 in *C. neoformans*, we constructed Cdh1-related strains. The target gene *CDH1* was replaced with the *NEO* marker gene using a Split Marker strategy. Subsequently, seven *cdh1*Δ mutant strains were obtained and confirmed via PCR and Southern blot. The plasmid for *CDH1* gene complementation, pTBL404, was transformed into the *cdh1*Δ mutant strain using biolistic transformation. Additionally, to obtain the *CDH1*^OE^ overexpression strain, i.e., the constructed *CDH1*^OE^ overexpression vector pTBL443 in which the expression of the *CDH1* controlled by actin promoter was transformed into the *cdh1*Δ mutant strain by biolistic transformation. The expression of the *CDH1* gene in transformants was confirmed through resistance screening, Western blot, and qRT-PCR analysis. As shown in Figure 3A,B, the Cdh1-HA protein size was consistent with expectations, and the RNA expression level was significantly higher compared to that of the wild-type H99 strain, thus successfully yielding the *CDH1*^OE^ overexpression strain.

### 3.4. Cdh1 Affects the Production of Virulence Factors of C. neoformans

The ability to grow at 37 °C, form melanin, and produce a polysaccharide capsule represent the three main virulence factors of *C. neoformans*, crucial for its survival in infected hosts and the initiation of cryptococcosis. To investigate the role of Cdh1 in the production of these virulence factors, we incubated *cdh1*Δ mutant, *cdh1*Δ::*CDH1* complementary, and *CDH1*^OE^ overexpression strains at 37 °C, plated them on a Niger seed medium to induce melanin formation, and inoculated them in a SAB + MOPS medium to induce capsule growth in order to observe the formation of virulence factors. The results of the capsule formation of the strain are shown in Figure 3C,D. Compared to the wild-type strain H99, the capsule thickness of the *cdh1*Δ mutant strain was significantly reduced. Similar results were observed with the overexpression of *CDH1*, indicating the importance of Cdh1 in the capsule synthesis of *C. neoformans* and its involvement in the process of capsule formation. Additionally, we examined the effects of the *cdh1*Δ mutant and *CDH1*^OE^ overexpression strains on melanin formation, and the results showed that the deletion and overexpression of the *CDH1* gene did not affect melanin formation in *C. neoformans* (Figure 3C).

To evaluate the role of Cdh1 on the growth of *C. neoformans* under different stress conditions, the series-diluted strains were then plated on YPD agar plates and incubated at 37 °C for two days for observation. After incubation, we found that the *cdh1*Δ mutant strain grew slowly at 37 °C (Figure 3E), indicating that Cdh1 is involved in maintaining the growth of *C. neoformans* at 37 °C. Meanwhile, we also noticed a slight growth defect of the *cdh1*Δ mutant in YPD with 1.5M KCl or 1.5M NaCl, indicating that Cdh1 is important for *C. neoformans* to respond to high-salt environments. Concurrently, we assessed the growth of the *cdh1*Δ mutant in a liquid YPD medium. The results revealed that the growth rate of the *cdh1*Δ mutant at 30 °C was comparable to that of other strains. However (Figure 3F), at 37 °C, its growth rate was significantly diminished relative to the other strains (Figure 3G). These findings suggest that the disruption of the *CDH1* gene affects capsule formation and growth at 37 °C or in high-salt environments, thereby indicating a potential role for Cdh1 in the pathogenicity of *C. neoformans*.

### 3.5. Cdh1 Regulates Fungal Virulence in C. neoformans

To explore the effect of Cdh1 on fungal virulence, C57 BL/6 mice were intranasally infected with the strains to assess the role of Cdh1 in the virulence of *C. neoformans*. Statistical analysis revealed that the mice infected with the wild-type H99 strain succumbed to the infection 19 days post-infection(dpi), with all of them dying within 28 days. In contrast, mice infected with the *cdh1*Δ mutant strain began to die at 30 dpi and did not all succumb until 80 dpi, signifying a significant decrease in pathogenicity. When infected with the *cdh1*Δ::*CDH1* complemented strain, the pathogenicity of the strain was not significantly different from that of the wild-type H99 strain. The mice began to die at 21 dpi, with all deaths occurring within 26 days. Following infection with the *CDH1*^OE^ overexpression strain, the mice started to die at 37 dpi, and all died within 55 days (Figure 4A).

To understand why the *cdh1*Δ mutant and *CDH1*^OE^ overexpression strains showed decreased virulence, we counted the fungal burden in the infected mice at the end of the animal experiment. The results showed no significant difference in the CFU statistics of different tissues of mice infected with *cdh1*Δ mutant, *cdh1*Δ::*CDH1* complemented, and *CDH1*^OE^ overexpression strains compared to mice inoculated with the H99 strain (Figure 4B). Pathological tissue sections revealed that infection with *cdh1*Δ mutant, *cdh1*Δ::*CDH1* complemented, and *CDH1*^OE^ overexpression strains resulted in severe damage to the main infected organs of mice, including lungs, brains, and spleens, with noticeable tissue lesions and a large number of yeast cells (Figure 4C, black arrow). This result was consistent with the tissue CFU statistics. Based on the fungal burden statistics and pathological section results of dead mice at the end of the experiment, it was evident that *C. neoformans* could accumulate in large quantities in the body after infection with *cdh1*Δ mutant and *CDH1*^OE^ overexpression strains, ultimately leading to the death of the mice. However, it was observed from the survival curves that infection with the *cdh1*Δ mutant and the *CDH1*^OE^ overexpression strains prolonged the survival days of mice and significantly decreased pathogenicity.

The markedly reduced growth rate observed in the *cdh1*Δ mutant compared to the wild-type strain at 37 °C indicates that the diminished pathogenicity of the *cdh1*Δ mutant during in vivo infection may be primarily attributable to its impaired growth at 37 °C. This finding complicates the effort to delineate the loss of virulence from thermosensitive effects. To further investigate this phenomenon, we conducted a pathogenicity assay utilizing a wax moth larvae model. The results demonstrated a significant attenuation of virulence in the *cdh1*Δ mutant at both 30 °C (Figure 4D) and 37 °C (Figure 4E), suggesting that the observed loss of pathogenicity in the *cdh1*Δ mutant is not merely a consequence of reduced heat resistance.

### 3.6. Cdh1 Regulates the Progression of Fungal Infection

The function of Cdh1 in *C. neoformans* infection was investigated through the quantification of fungal burden and histopathological analysis of lung, brain, and spleen tissues at 7, 14, and 21 dpi with the H99 strain, *cdh1*Δ mutant, and *CDH1*^OE^ strains. In this study, we closely monitored the dynamic changes of *C. neoformans* within the host. Tissue fungal burden analysis indicated that yeast cells of *C. neoformans* were recovered from the tissues of mice infected with the H99 strain at 7 dpi, with the fungal burden progressively increasing over time (Figure 5A). Mice infected with the *cdh1*Δ mutant and *CDH1*^OE^ strains exhibited a similar trend in lung tissue but with a significantly lower fungal burden than the H99 strain. At 21 dpi, the *cdh1*Δ mutant and *CDH1*^OE^ strains began to spread to the brains, while no cryptococcal cells were isolated from the spleens. Histopathological sections revealed the presence of yeast cells in lung tissues at 7 dpi in all infected groups, with tissue damage increasing over time. However, infection with the *cdh1*Δ mutant strain or the *CDH1*^OE^ strain caused less tissue damage than the H99 strain (Figure 5B, left panel). At 21 dpi, lesions and a large number of yeast cells of *C. neoformans* were detected in the brains or spleens of mice infected with the H99 strain, whereas no significant lesions or yeast cells were found in the brains and spleens of mice infected with the *cdh1*Δ mutant and *CDH1*^OE^ strains until 21 dpi (Figure 5B, middle and right panels), indicating that the progression of the fungal infection is significantly delayed in mice infected with the *cdh1*Δ mutant and *CDH1*^OE^ strains. The findings from tissue sampling at different time points underscore the critical role of Cdh1 in the progression of *C. neoformans* infection (Figure 5D).

### 3.7. Cdh1 Affects the Proliferation of C. neoformans Inside Macrophages

The results of the fungal pathogenicity assay showed that Cdh1 affected the virulence of *C. neoformans*. To elucidate the reasons for the changes in the pathogenicity of *cdh1*Δ mutant and *CDH1*^OE^ strains in the host, the growth dynamics of *C. neoformans* cells after incubation with mouse serum for varying durations were examined to investigate the potential influence of host serum components on the growth of *C. neoformans*. The wild-type strain H99 served as the control, while the *cdh1*Δ mutant and *CDH1*^OE^ strains were co-incubated with mouse serum. Subsequently, the growth of each strain was assessed after 1, 2, 3, and 4 h of co-incubation. The findings showed that there was no significant difference in survival rates among the *cdh1*Δ mutant, *CDH1*^OE^, and wild-type strains after co-incubation with mouse serum. This suggests that host serum components did not exert a discernible impact on the growth of Cdh1-related strains (Figure 6A).

The pulmonary macrophages serve as the first line of defense against *C. neoformans*. Therefore, we investigated how *CDH1*-related strains interact with J774 macrophages from mice. There were no significant differences between the *cdh1*∆ mutant, *CDH1*^OE^ strain, and the wild-type H99 strain after 2 and 4 h of co-incubation with macrophages. However, after 24 h of incubation, the proliferation of the *cdh1*∆ mutant and *CDH1*^OE^ strains did not increase as much as the wild-type H99 strain and the *cdh1*Δ:: *CDH1* complemented strain but instead remained at a proliferation level comparable to that observed after 4 h of incubation (Figure 6B). These findings indicate that the reduced pathogenicity of the *cdh1*Δ mutant and *CDH1*^OE^ strains results from their limited proliferation within host macrophages.

### 3.8. Dynamic Changes of Chemokines in Lung Tissues of Mice Infected with Cryptococcus Strains

Chemokines are crucial in generating adaptive immune responses and promoting the pathogenesis of various diseases. Compared to the H99 strain infection, Cxcl1, which can recruit and activate neutrophils, and Ccl17, secreted by M2 macrophages to act on Th2 cells, showed a gradually increasing expression level in the lungs of mice infected with the *cdh1*Δ mutant strain. Notably, after 14 and 21 days of infection, the expression levels of *CXCL1* and *CCL17* significantly increased compared to the H99 infection group (Figure 7). Conversely, following infection with the *CDH1*^OE^ strain, the expression of *CXCL1* in mouse lungs exhibited a gradual decline, ultimately reaching levels that were significantly lower than those observed in the H99 infection group by day 21. Furthermore, Ccl20, which plays a crucial role in the migration of Th17 cells to the site of inflammation, showed no significant difference in expression in the lungs of mice infected with the *cdh1*Δ mutant strain 7 and 14 dpi. However, a marked increase in the expression of *CCL20* was noted 21 dpi with the *cdh1*Δ mutant strain (Figure 7). In the early stage of infection (7 dpi), both the *cdh1*Δ mutant and *CDH1*^OE^ strain infections were shown to stimulate lung tissue to secrete greater levels of the Th1 chemokine Cxcl10, which is known for its direct killing activity against *C. neoformans* [49]. Nevertheless, the expression level of *CXCL10* exhibited a significant reduction in subsequent stages (14 and 21 dpi) (Figure 7). Moreover, Ccl2, a chemokine critical for the promotion of monocyte/macrophage migration and infiltration to infection sites, displayed significantly lower expression levels in the lungs of mice infected with *cdh1*Δ mutant and *CDH1*^OE^ strains 14 and 21 dpi, in contrast to the H99 group. The expression levels of *CXCL1*, *CCL17*, and *CCL20* were significantly upregulated 21 days after infection with the *cdh1*Δ mutant strain, indicating a pronounced activation of lung tissue recruitment of neutrophils and enhanced Th2 and Th17 immune responses (Figure 7). Additionally, the elevated expression of *CCL20* across all time points following *CDH1*^OE^ strain infection suggests that the Th17 immune response is more robustly activated in this context. The observed low levels of *CCL2* imply that infections with the *cdh1*Δ mutant and *CDH1*^OE^ strains exert a limited effect on macrophage activation, resulting in fewer monocytes/macrophages migrating to the lungs.

## 4. Discussion

*C. neoformans* is a critical fungal pathogen that threatens public health worldwide. Due to the limited availability of antifungal drugs and the emergence of drug-resistant strains with toxic side effects, it is crucial to explore effective treatments for cryptococcosis. Researching the pathogenesis of *C. neoformans* is important for better understanding the fungus, identifying new antifungal drug targets, and developing more effective antifungal drugs. To investigate the role of the Cdh1 protein in the pathogenicity of *C. neoformans*, we created the *CDH1* gene disruption, complementation, overexpression, and GFP-Cdh1 fusion expression strains using biolistic transformation. Subcellular localization studies revealed that the Cdh1 protein is located in the nucleus of *C. neoformans*. The *cdh1*Δ mutant strain showed a growth defect at 37 °C. Both *CDH1* deletion and overexpression significantly reduced the capsule thickness of *C. neoformans*, suggesting that Cdh1 may play a role in regulating capsule formation. Furthermore, our results from the murine inhalation model of the systemic infection of cryptococcosis also indicate that Cdh1 plays a significant role in the pathogenicity of *C. neoformans*.

In *C. neoformans*, CNAG_03191 was annotated as a putative protein. Since its sequence has the highest sequence similarity as that of *S. cerevisiae* Cdc20 homolog 1 (Cdh1) and both contain the WD40 domain, it was also named Cdh1 in this paper. Repetitive WD domain proteins are a large family in all eukaryotes and are involved in various processes, such as signal transduction, transcriptional regulation, cell cycle control, and apoptosis [48]. In mitotic cells, the activity of the Cdh1 protein shows cellular periodicity; Cdh1 maintains a low phosphorylation level in the G1 phase and mitosis anaphase and a high phosphorylation level in the G2 phase, the S phase, and early mitosis [50]. Cdh1 is maintained in a repressive state at mitosis telophase and in the G1 phase to ensure that cells undergo precise DNA replication during the S phase. In the late stage of mitosis, the anaphase-promoting complex (APC) can be activated by cell division cycle 20 homolog (Cdc20) or Cdh1, which degrades cell cycle proteins through ubiquitination, thereby inactivating the corresponding CDK and completing a cell cycle [51]. In vertebrates, the loss of Cdh1 results in the accumulation of mitotic cell cycle proteins cyclin A and cyclin B in the G1 phase and a delay in mitotic exit [50]. As the role of Cdh1 in the cell cycle of *C. neoformans* is still unknown, the function of the *CDH1* gene in *C. neoformans* can be fully elucidated by examining the cell cycle progression of the *cdh1*∆ mutant strain.

In budding yeast, Cdh1 mediates the degradation of APC/C substrates, and the Cdc20/Cdh1 protein family can be used as specific activators of downstream substrates of APC/C [52,53]. This conclusion was also confirmed by Gieffers et al., who proposed that Cdh1 is the activating subunit of APC/C [54]. Cdc20 and Cdh1 function at different cell cycle stages and contribute to the substrate specificity of APC/C. Cdc20 activates APC in early mitosis, while Cdh1 activates APC in G1 and mitosis anaphase [55]. As a target gene of Grp1, the downstream substrate of F-box protein Fbp1 (data not published), whether Cdh1 also has the function of specifically activating Grp1, the downstream substrate of the E3 ubiquitin ligase SCF complex, remains to be further explored.

The capsule is the main virulence factor, which can inhibit the phagocytic activity of host immune cells towards *C. neoformans*. Mutant strains with a damaged capsule structure are often more susceptible to phagocytosis by macrophages [10,11,12], and an increase in thickness reduces phagocytic efficiency, making it difficult for cells to break through the blood–brain barrier [8,9], indicating that the normal formation of capsules is closely related to virulence. The deletion of the *CDH1* gene significantly reduced the capsule thickness of *C. neoformans*, and overexpression of this gene also showed similar results (Figure 3C,D), indicating that the normal expression of *CDH1* plays an important role in capsule synthesis and may be involved in regulating the process of capsule formation. The performance of Cdh1 in capsule synthesis and growth at 37 °C also indirectly suggests its possible association with pathogenicity.

The intranasal inhalation experiment conducted in mice demonstrated that infection with the *cdh1*Δ mutant and *CDH1*^OE^ strain led to a notable extension in the survival time of the mice and a reduction in the pathogenicity of the strains (Figure 4A). In addition, the assessment of tissue fungal burden and observation of tissue lesions at specific time points for mice inoculated with *cdh1*Δ mutant and *CDH1*^OE^ strains revealed a gradual increase in the number of *C. neoformans* in tissues throughout the infection, albeit at significantly lower levels compared to the wild-type strain (Figure 5A). The infection kinetics of the *cdh1*Δ mutant and *CDH1*^OE^ strains were found to be akin to those of the wild-type H99 strain, although they were significantly delayed. The growth rate of the *cdh1*Δ mutant was observed to be significantly lower than that of the H99 strain at 37 °C, as illustrated in Figure 3G. Together with the pathogenicity assay outcomes conducted using *Galleria mellonella* larvae, depicted in Figure 4D,E, these findings suggest that the diminished pathogenicity of the *cdh1*Δ mutant strain primarily arises from impaired thermotolerance. Conversely, the *CDH1*^OE^ strain presents a more intricate mechanism of action, exhibiting normal growth at 37 °C while simultaneously demonstrating a pronounced reduction in virulence. This phenomenon necessitates further investigation to elucidate the underlying mechanisms.

Human respiratory tract infection occurs by inhaling dried spores or yeast cells of *C. neoformans* in the environment. Host immune cells, including macrophages and dendritic cells, are the first to interact with invading pathogens and participate in the early recognition and clearance of infection [56]. Here, the interaction between *C. neoformans* and macrophages was examined to explore the reasons for the changes in the pathogenicity of Cdh1-related strains in the host. When the Cdh1-related strains were co-incubated with mice macrophages, the *cdh1*∆ mutant strain showed a significant decrease in proliferation at 24 h of incubation (Figure 6B), which was consistent with the slower growth of the strain at 37 °C when the strain was tested for virulence factors (Figure 3E), indicating that the growth of the strain itself caused the reduced proliferation of the *cdh1*∆ mutant strain in macrophages.

Exploring the immune response during the interaction between hosts and pathogens can help us better understand the molecular mechanisms of pathogen invasion in hosts and provide ideas for analyzing pathogenicity and finding therapeutic targets. According to qRT-PCR analysis of chemotactic factors in mice lung tissue, infection with the *cdh1*Δ mutant strain can activate the recruitment of neutrophils in lung tissue, as well as Th2 and Th17 immune responses, to a greater extent. At the same time, its effect on macrophage activation is limited (Figure 7). The chemokine Cxcl1 recruits and activates neutrophils to combat infections and kills microorganisms in tissues [57,58]. Although neutrophils contribute to resistance to *C. neoformans* infection, it is unnecessary for the host to clear *C. neoformans* [59,60]. Cxcl10, belonging to Th1 chemokines, can target the cell wall/membrane and intracellular structures of *C. neoformans* and have direct killing activity against *C. neoformans* in vitro [49]. In the early stage of infection (7d), infection with the *cdh1*Δ mutant strain can stimulate more robust secretion of Cxcl10 in lung tissue, but in the later stage, the expression level of *CXCL10* is significantly reduced, which may be the reason why infection with the *cdh1*Δ mutant strain cannot effectively be eliminated from the body of the mice.

Currently, antifungal drugs used to treat cryptococcosis include azoles, polyenes, and pyrimidine analogs. The azoles specifically target the ergosterol biosynthetic enzyme Erg11, while polyenes directly target ergosterol, and pyrimidine analogs block the synthesis of DNA and RNA [61]. In the future, we can explore the sensitivity of the *cdh1*Δ mutant strain to various antifungal drugs, which may lead to the discovery of new therapeutic targets and facilitate the development of improved antifungal medications.

As a basidiomycete with a bipolar mating system, *C. neoformans* has a chromosomal region with a mating type locus, the MAT locus, which corresponds to two mating types: α and **a**. Strains with different mating types can carry out cell fusion, hyphal growth, and basidiospore production under mating induction conditions [62,63]. Basidiospores produced through the sexual reproduction process are resistant to external stress environments and are important pathogens that can cause fatal cryptococcal meningitis [64]. Wu et al. showed that the deletion of *CDC4*, a cell cyclin-related protein, could not undergo nuclear division during mating, blocking meiosis, thus affecting the sexual reproduction of *C. neoformans* [65]. The mutant strain lacking the cell cycle-related protein Cbc1 lost the ability to produce basidiospores [33]. These results suggest that the cell cycle-related proteins of *C. neoformans* play an essential role in producing infectious basidiospores. In this study, we mainly explored the effect of Cdh1 on the pathogenicity of *C. neoformans* but did not study its role in sexual reproduction. In the future, the molecular mechanism of Cdh1 in the mating sporulation can be further analyzed to provide biological data for the morphological development of *C. neoformans*.

## Figures and Tables

**Figure 1 jof-10-00891-f001:**
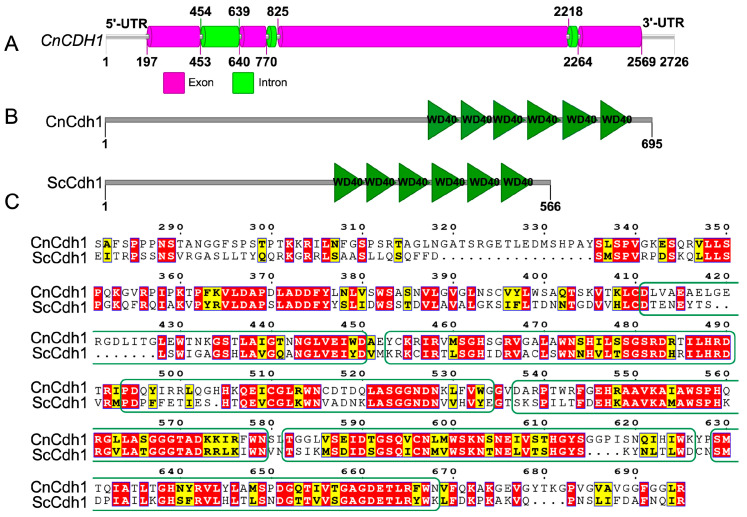
Sequence analysis of Cdh1. (**A**) Sequence structure of the *CDH1* gene; (**B**) Comparison of protein structures between *C. neoformans* and *S. cerevisiae* Cdh1; and (**C**) *C. neoformans* Cdh1 protein shows high similarity to the Cdh1 proteins of *S. cerevisiae*. The WD40 domains were labeled by a green rectangular box.

**Figure 2 jof-10-00891-f002:**
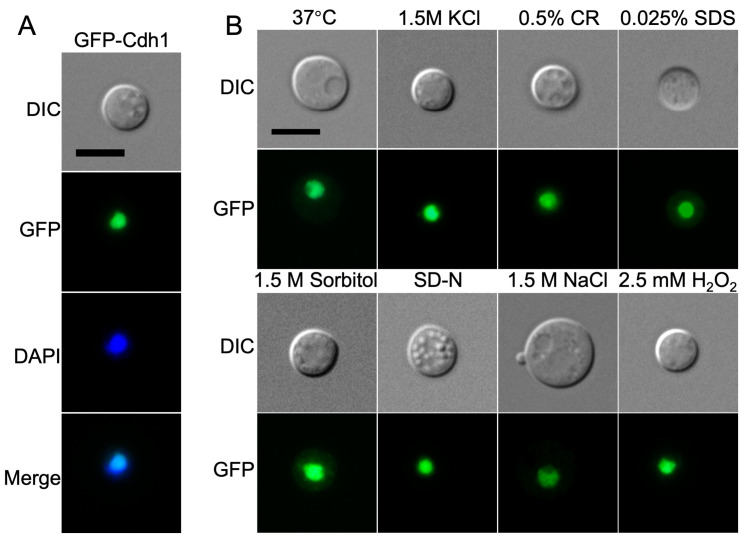
Cdh1 is localized in the nucleus of *C. neoformans*. (**A**) The GFP-Cdh1 fusion protein was targeted to the nucleus of *C. neoformans*. The nuclei of the GFP-Cdh1 strain were stained with 10 µg/mL DAPI solution, and the localization of GFP-Cdh1 protein in *C. neoformans* was observed by fluorescence microscopy (Axo observer 3; Zeiss, Oberkochen, Germany). (**B**) Subcellular localization of the GFP-Cdh1 fusion protein under different stress conditions. CR, Congo red; SD-N, Synthetic Dropout medium without nitrogen; and DIC, differential interference contrast. Scale bar: 5 μm.

**Figure 3 jof-10-00891-f003:**
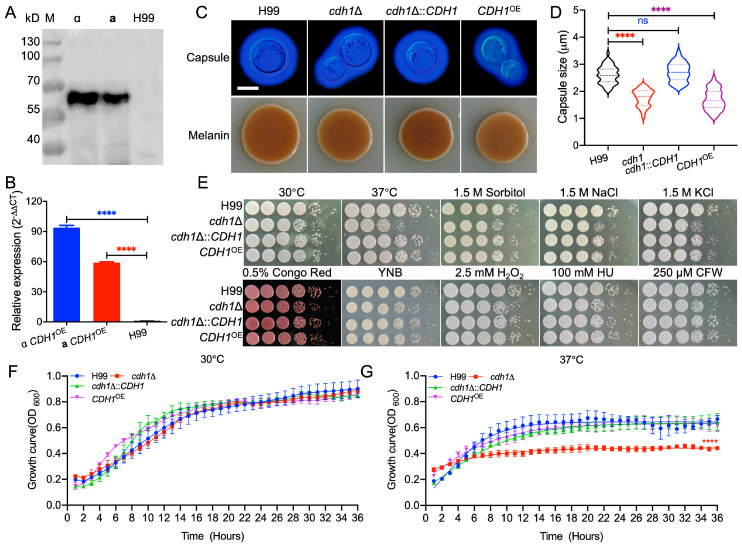
Cdh1 affects the development of the classical virulence factors and growth of *C. neoformans*. (**A**) Validation of the *CDH1*^OE^ overexpression strains by Western blot. α: α mating type *CDH1*^OE^ strains; **a**: **a** mating type *CDH1*^OE^ strains. (**B**) Measurement of gene expression levels of the *CDH1*^OE^ overexpression strains via relative qRT-PCR analysis. M: PageRuler prestained protein ladder. ****, *p* < 0.0001. (**C**) Observation of capsule formation and melanin production. Strains were inoculated in an SAB medium diluted with MOPS, and capsule production was visualized by ink staining after induction at 37 °C for 1 day. Melanin levels produced by each *C. neoformans* strain were imaged after incubation on Niger seed plates for 3 days at 37 °C. Scale bar: 5 μm. (**D**) Statistical analysis of the capsule formation size. ns, not significant and ****, *p* < 0.0001. (**E**) Growth of *cdh1*Δ mutant and *CDH1*^OE^ overexpression strains under different stress conditions. Strains grown overnight were diluted in a ten-fold series and plated on YPD agar plates supplemented with different stressors and incubated at 30 °C for 2–4 days. Incubation conditions are marked at the top, and the *C. neoformans* strains are marked on the left. ns: not significant and ****, *p* < 0.0001. Scale bar: 5 μm. *C. neoformans* cell growth was assayed in a BMG LABTECH multifunctional enzyme labeling instrument (FLUOstar Omega, Germany). Briefly, 100 µL of inoculum (1 × 10^7^ cells/mL) was incubated at 30 °C (**F**) or 37 °C (**G**) with shaking (200 rpm), and OD600 was measured in real time every two hours. Each experiment was performed in triplicate Error bars indicate standard deviations. The asterisk indicates the *cdh1*Δ mutant maximal growth compared to the wild-type H99 strain. ****, *p* < 0.0001.

**Figure 4 jof-10-00891-f004:**
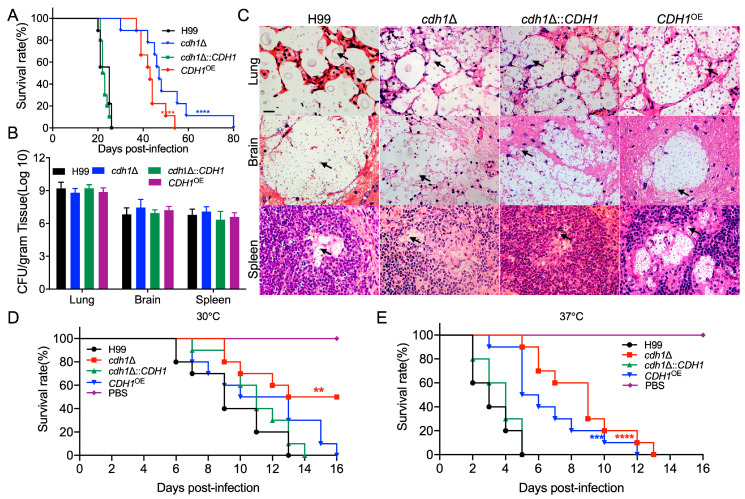
The role of Cdh1 in the pathogenicity of *C. neoformans*. (**A**) Survival curves of C57 BL/6 mice infected with H99, *cdh1*Δ mutant, *cdh1*Δ::*CDH1* complemented, and *CDH1*^OE^ strains (n = 10 per group). ****, *p* < 0.0001. (**B**) The number of cryptococci recovered from lungs, brains, and spleens of mice infected with H99, *cdh1*Δ mutant, *cdh1*Δ::*CDH1* complemented, and *CDH1*^OE^ strains at the end of the experiment (n = 5 per group). (**C**) Histopathological slides stained with H&E of lungs, brains, and spleens from mice infected with H99, *cdh1*Δ mutant, *cdh1*Δ::*CDH1* complemented, and *CDH1*^OE^ strains were visualized by light microscopy. The cryptococcal cells are indicated by arrows. Scale bar: 20 μm. Survival curves of *Galleria mellonella* infected with H99, *cdh1*Δ mutant, *cdh1*Δ::*CDH1* complemented, and *CDH1*^OE^ strains (n = 10 per group) at 30 °C (**D**) or 37 °C (**E**). **, *p* < 0.01; ***, *p* < 0.001; and ****, *p* < 0.0001.

**Figure 5 jof-10-00891-f005:**
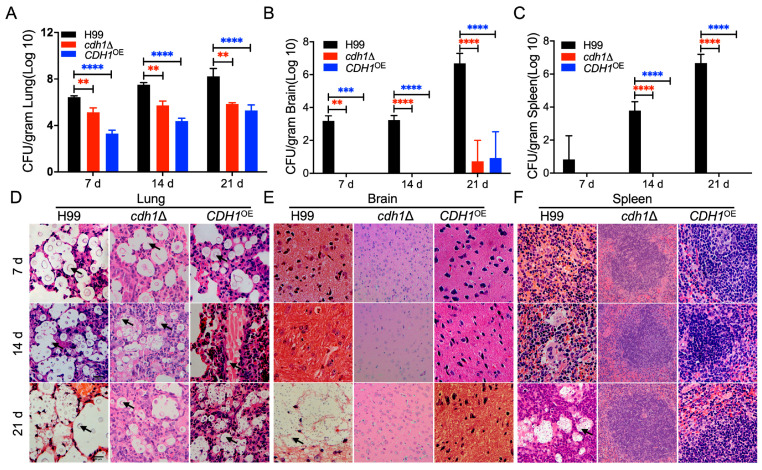
Fungal infection progression of the *cdh1*Δ mutant and *CDH1*^OE^ strain in mice. Statistical analysis of fungal burden in the lungs (**A**), brains (**B**), and spleens (**C**) of mice infected by each cryptococcal strain. Data are presented as means ± SD of three mice. **, *p* < 0.01, ***, *p* < 0.001, and ****, *p* < 0.0001. H&E-stained slides of infected lungs (**D**), brains (**E**), spleens (**F**), and visualized by light microscopy. The cryptococcal cells are indicated by arrows. Scale bar: 20 μm.

**Figure 6 jof-10-00891-f006:**
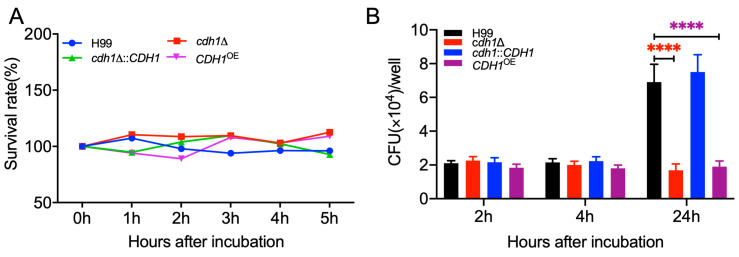
Cdh1 plays an important role in the survival of *C. neoformans* inside macrophages. (**A**) Survival rate of *C. neoformans* after co-culturing with mouse serum at 37 °C. Overnight cultures of each *C. neoformans* strain were co-cultured with mouse serum, and after different incubation times, 100 μL of diluted suspension was plated on the YPD agar plates to count the number of the *C. neoformans* colonies. (**B**) Proliferation of *C. neoformans* in macrophages. After incubation with macrophages for different lengths of time, 100 μL of diluted lysates of the J774 macrophages was plated on YPD agar plates to count the number of *C. neoformans* within the macrophages. Significance was determined using a one-way ANOVA with Tukey’s multiple comparisons in GraphPad Prism 8.0. ****, *p* < 0.0001.

**Figure 7 jof-10-00891-f007:**
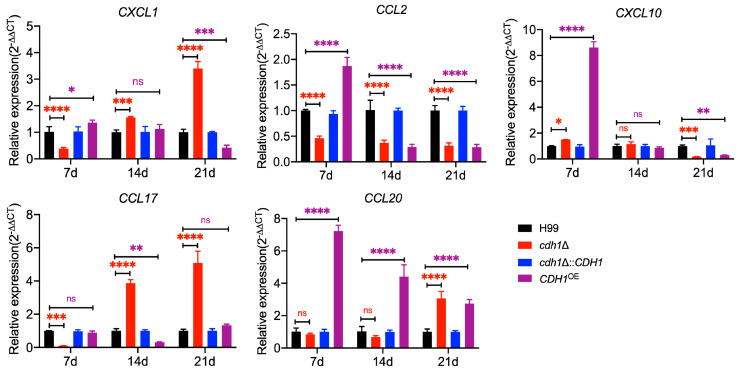
Analysis of expression levels of chemokine genes in lung tissue during infection by the *cdh1*Δ mutant. The significance for each time point was determined using a two-way ANOVA with Dunnett’s multiple comparison test in GraphPad Prism 8.0. ns: not significant; *, *p* < 0.1; **, *p* < 0.01; ***, *p* < 0.001; and ****, *p* < 0.0001.

## Data Availability

The datasets presented in this study can be found in online repositories. The names of the repository/repositories and accession number(s) can be found in the article/Appendix A.

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
