# Peer review of "Role of the Anaphase-Promoting Complex Activator Cdh1 in the Virulence of Cryptococcus neoformans"

_jof, 2024, doi:10.3390/jof10120891_

Round 1
Reviewer 1 Report
This manuscript studies a cell cycle control protein of C. neoformans. This protein has not been studied before, so knowledge of its role adds to the corpus of our understanding of this pathogen.
Some of the findings are interesting, especially regarding the phagocytosis rate of the knockout and overexpression strains. However, the phagocytosis experiments need to be better explained, and some issues of data interpretation remain. Ditto for the fungal burden experiments.
45-6: "conversely", not "on the contrary"
51-2: unclear sentence
68: "cell cycle-related", not "cell cyclin-related"
71: "intricate structure" is a non-descriptor, and the reasons for the difficulty in treating fungal infection are complex and varied.
85: LB actually means "lysogeny broth", not "Luria-Bertani", and thus you can drop "media", since it's redundant with "broth".
88: Sabouraud, with capital "S"
Section 2.3: when describing microscopy experiments, always include the information on the microscope objectives use, including the numerical aperture. Also, the addition of Triton X-100 isn't a reaction, replace the word with "permeabilization".
Section 2.4: again, indicate which objective was used.
Section 2.6: the two paragraphs state the same procedures, the second one with a little more detail.
Line 221: you should indicate which macrophages were used, and when were interferon gamma and LPS added, and for how long. Only much later, when discussing figure 6, does the reader find out you've used J774.
Section 2.7 as a whole: what you describe isn't (just) a phagocytosis assay. It's a post-phagocytosis yeast CFU assay, also known as a macrophage fungicidal activity assay. Sure, you can use the zero and two-hour time points to estimate the efficiency of phagocytosis per strain, since you know how many yeast cells you've added to each well, and the CFU counts tell you how many you've recovered from within the cells. However, lines 221-6 aren't clear how you've performed the experiment: you say you incubated for two hours before washing the unphagocytosed cells, but you don't say that time zero is. Is time zero immediately after two hours of incubation and washing, or even before incubation? I ask because "22 hours" is a peculiar time point, but it makes sense if you're counting from the washing, because then it's actually 24 hours.
This is a crucial issue. If you're calling zero hour the time point immediately after washing the unphagocytosed cells, then that is the time point that indicates the efficiency of phagocytosis: two hours is the peak of phagocytosis and before any yeast cells have had time to divide, die, suffer vomocytosis or lyse their macrophages. But by that logic, the two- and 22-hour time points don't reflect just phagocytosis, but the effects of all these other processes. You should then use the zero-hour time point to normalise the CFU results of the latter points in order to compare CFUs among the strains tested.
I also note that you removed the culture medium, lyse the cells and plate only the lysate. This means you've discarded with the medium the yeast cells that suffered vomocytosis or that lysed their macrophages. So you've only performed a CFU assay on cells that were still inside macrophages. If there are any differences in lytic or non-lytic exocytosis across the strains, you've missed them, and in fact cannot state that the observed differences reflects the fungicidal activity of the macrophage against them.
Section 3.1 and figure 1: please state what WD-40 stands for and why that is relevant for identifying the protein. I know you talk about it in the discussion, but you need to at least briefly state it the first time the name appears.
Section 3.2 and back to section 2.2, lines 134-47: you never state which promoter you have used for overexpressing Cdh1. Even if there's a plasmid map in the supplements, you must state this piece of information in the text. And plasmid pTBL443 has been reported before, please add the reference.
Figure 3B and related text: a conceptual problem here is that you're making your comparisons against H99, but you state in the methods that the parental strains for deletion and overexpression are the mating types a and alpha. H99 is an alpha strain, but there are at least two laboratory-derived subclusters of H99 strains, "stud" and "wimp", which were generated by replating across the years, and they behave very differently in terms of sexual reproduction and virulence. Regardless, the good practice is to always compare to the parental strains, which I imagine were KN99a and KN99alpha in this case. But since the fold-changes in figure 3B are quite high, I think it's fair to say that overexpression was achieved.
Figure 2 and related text: I don't think anybody doubts that a Cdh1 homologue with 75% similarity to the baker's yeast protein will localise to the nucleus, but rigorously speaking, you should have transformed the knockout strain instead of H99 and shown not only nuclear localisation, but that the chimaeric protein retains its function. However, the observation stands on grounds of plausibility.
Stress agents in figures 2 and 3: you haven't explained the spot-dilution assay in the methods, and you haven't spelled out the initials of the stressors anywhere. I imagine HU is hydroxyurea, but I failed to account for SD-N.
About incubation at 37 °C, I think you should've complemented that with a growth curve in liquid medium so you could quantify the magnitude of the effect. Spot dilution assays are at best semiquantitative, and the difference you've seen is small. And you've written figure 4E on line 307, but that's figure 3E. By the way, you don't mention it anywhere, but it's clear that 1.5 M KCl or NaCl also inhibit the mutant as much as incubation at 37 °C. If you discuss one small effect, you may as well discuss all of them.
Section 3.5, figure 4: because the mutant strain has a small sensitivity to growth at 37 °C, the in vivo infection data could be entirely due to this effect. It's difficult to separate loss of virulence independent of thermosensitivity in mammalian pathogens, but in the case of cryptococci, you could've compared survival of wax moth larvae at 37 °C and 30 °C. If at 30 °C virulence of the mutant were similar to that of the parental strain, then losses of virulence at 37 °C could be attributable to loss of thermotolerance. This is especially important because the overexpression strain behaved similarly to the knockout strain, but while it failed to restore capsule growth, it did apparently restore thermotolerance. This is another reason why growth curves in liquid medium would've helped quantifying the role of loss of thermotolerance for the mutant. I'll also note that mutant strain exhibited a sizeable capsule in the tissues, which suggests that capsule production isn't severely affected in the mutant in vivo.
Also, of relevance to the next section: what do you call "the end of the experiment" on figure 4B and related text? Do you mean when the mice had reached the euthanisation point?
Figure 6 and related text: figure 6B is a conundrum. These are cells opsonised with serum, and the overexpression strain is normocapsular. You show that the overexpression mutant was phagocytosed at less than 10% of the other strains. I'm sorry, but extraordinary observations need substantial backing. Have you ruled out experimental mistakes? Do the scatter bars reflect multiple experiments performed independently, or technical replicates? Have you attempted antibody opsonisation to confirm this? It makes simply no sense that the overexpression strain is phagocytosed at such low levels, and yet it kills more quickly than the mutant strain in the in vitro model. Other than that, the delay in proliferation seen in the mutant is entirely consistent with a loss of thermotolerance. Thus, performing a survival assay using wax moths at 30 °C becomes even more important to tease out the role of Cdh1 loss on virulence apart from loss of thermotolerance.
Section 3.8 and figure 7: why did you not perform the chemokine PCR with the complemented and overexpression strains as well?
On line 431, you need to put the reference for the killing activity of CXCL10 against cryptococci, not just in the discussion.
I also recommend that you pass a fine comb on gene terminology. In eukaryotes: all-caps, italic for genes, lowercase italics for mutants and regular, uppercase first letter for proteins. I've seen a couple mistakes, like cdh1 in lowercase regular on line 473.
482-5: missing reference. In fact, I couldn't find a reference on Grp1 being a target of SCF or Fbp1. Please clarify.
Lines 505-6: that's a bit of a stretch in the case of the overexpression strain at least. If your phagocytosis data are true, then the overexpression strain may be hypovirulent because it has reduced access to the Trojan Horse route of cryptococcal dissemination. And because you didn't perform the fungal burden time curves in figure 5 with the overexpression strain, you can't say anything about that, either.
Lines 517-26: the logic here is faulty. The overexpression strain has a capsule of similar thickness as that of the mutant strain, as per figure 3D. And yet, the mutant strain has no phagocytosis defect? And in fact, you have the causal nexus wrong: thicker capsules are know to reduce phagocytosis, that's one of the primary roles of GXM in virulence. Hypocapsular strains normally are more phagocytosed, not less. And in any case, this is only valid for nonopsonised cells, and here you've used serum to mediate phagocytosis. As I said before, you need to investigate the findings with the overexpression strain more closely in order to be able to state anything.
Lines 526-30: this is a more promising line of inquiry. Sure, if the overexpression strain produces more App1, that could explain the effect in principle. However, by that logic the knockout strain would be expected to produce less App1 and be taken up more avidly by macrophages. Also, the effect of App1 is at most a 50% reduction in phagocytosis, here you've seen a 90% reduction. You should at least perform a quantitative PCR of App1 comparing the parental, knockout and overexpression strains before you can say anything.
Line 541: truncated sentence - "can target the cell wall/membrane and intracellular of C. neoformans and have" - intracellular what?
Line 545: the infection doesn't eliminate, it is eliminated. Reword this sentence.
Line 548: add "synthesis" before "inhibitors".
Lines 548-9: only polyenes and azoles work like that.
Reviewer 2 Report
A more detailed description of some methods is missing.
What lenghts are required for homologous recombination in C. neoformans?
Have you noticed any differences between the strains with different mating types?
Line 116 add „and“
Line 160 what was the pH of the PBS buffer?
Lines 206-214 this section is written twice
Line 281 recommend to mention of the stress conditions in the section Matherial and Methods 2.3.
Fig E C,D,E indicate if a or α strains were used
Section 2.7. line 219 continue to describe the method with mice serum as used for Fig. 6A
Line 542 „in vitro“ should be italicised
Line 548 is „ergosterol on membranes“ a correct term? Perhaps “in“ should be used.
Round 2
Reviewer 1 Report
This version of the manuscript is much improved.
However, given that you're still performing new experiments to address my concerns in the review of the first version, I'll withhold judgement on most of the points, lest you have to waste time rereading a review addressing the same points. Just know that I expect to see all points for which you're performing new experiments addressed in the next version. Good luck with them.
As for points that I can address in this version, I comment on the next section.
I'm recommend this paper for rejection because this seems to be the only option that will cause the editors to give you more time to finish the new experiments. It's not because I think the paper is worse than before, on the contrary.
An important point is statistics.
Figure 3G (which in your response letter you call figure S2) does not indicate how the statistical analysis was performed. For one thing, I see no scatter bars, so how many biological replicates of the growth curves have you performed? For another, the difference in growth rate of the mutant at 37 °C is very small, perhaps because you terminated the measurements early, before the strains reached the stationary phase. You should do a longer culture and compare the curves by growth curve fitting and linear regression statistics. The Gomperz model normally works well for cryptococci.
Figure 7 needs to avoid the problem of multiple comparisons. You generate lots of p values, which increases the chance of some being considered significant by chance. Please indicate in the figure legend which test was used.
Reviewer 2 Report
The authors have answered all of my questions.
The paper has been improved.
Author Response
Thank you for taking the time and effort to review my article. I am grateful for your support, and I deeply appreciate your dedication to the academic community.
Round 3
Reviewer 1 Report
This manuscript is much improved. You have addressed most of my points of concern and the new experiments speak for themselves. I have minor edits to recommend, and maybe statistics can be improved, as detailed below.
Figure 3G and related text: you said in the response letter that you had analysed the growth data using regression and fitting the curves with the Gomperz equation, and yet you present the same inappropriate analysis using ANOVA as before. Once again: ANOVA mustn't be used to compare dependent data, even with a correction for multiple comparisons. Just do a curve fit and ask Prism to compare the growth rates or the maximal growth and it will give you the differences. It's not that I object to the finding (the mutant is clearly thermolabile), but you need to perform the comparison correctly.
Line 421: you wrote "30 °C (F)", but I'm pretty sure you meant "37 °C (E)".
Figure 6C. It's not clear from the comparison bar with the asterisk what it is that you're comparing, though I imagine it's the overexpression strain with the WT strain. Please improve the presentation to make it inequivocal. Perhaps placing an asterisk on top of the overexpression strain histogram and indicating in the legend that the comparison what against the WT strain.
Discussion: I think you need to stress the fact that much of the loss of virulence in the mutants is due to loss of thermotolerance. The Galleria experiment makes that clear, though naturally not all of the phenotypes are due to that, given that the mutant is hypovirulent. I also recommend indicating that the OE strain has a more complex phenotype, because it grows normally at 37 °C but shows a marked loss of virulence nonetheless.
And then there's figure 6B. You still haven't addressed my main concern regarding it: why is the overexpression strain phagocytosed at such a lower rate than the WT and knockout strains? Not only does it make no sense in light of its being nearly normocapsular, it cannot be explained by APP1 expression, because in figure 6C you show the OE strain also expresses less APP1 than the WT strain. I had suggested you redo the phagocytosis experiment with a different opsonin like the 18B7 mAb, but it seems this suggestion went unheeded.
